# Metabolomic Profiling and Bioanalysis of Chronic Myeloid Leukemia: Identifying Biomarkers for Treatment Response and Disease Monitoring

**DOI:** 10.3390/metabo15060376

**Published:** 2025-06-06

**Authors:** Selim Sayın, Murat Yıldırım, Batuhan Erdoğdu, Ozan Kaplan, Emine Koç, Tuba Bulduk, Melda Cömert, Mustafa Güney, Mustafa Çelebier, Meltem Aylı

**Affiliations:** 1Department of Hematology, Gülhane Training and Research Hospital, Ankara 06010, Türkiye; murat.yildirim@sbu.edu.tr (M.Y.); dr.batuhan@gmail.com (B.E.); tubakirazbulduk@gmail.com (T.B.); melda.comert@sbu.edu.tr (M.C.); ayli.meltem@gmail.com (M.A.); 2Department of Analytical Chemistry, Faculty of Pharmacy, Hacettepe University, Ankara 06230, Türkiye; ozankaplan@hacettepe.edu.tr (O.K.); emikocc25@gmail.com (E.K.); celebier@hacettepe.edu.tr (M.Ç.); 3Blood Center, Gülhane Training and Research Hospital, Ankara 06010, Türkiye; mustafa.guney@sbu.edu.tr

**Keywords:** chronic myeloid leukemia, metabolomics, molecular response, tyrosine kinase inhibitors, LC/MS

## Abstract

Background: Including Chronic Myeloid Leukemia (CML) patients with deep molecular responses (MR4.5) and those with suboptimal responses provides valuable insights into treatment-associated metabolic changes. This study aimed to characterize the metabolomic alterations associated with CML and identify potential biomarkers for treatment response, particularly in patients achieving a deeper molecular response versus those with poorer responses. Methods: Plasma samples were collected from 51 chronic-phase CML patients and 24 healthy controls. CML patients were classified into two groups based on molecular responses: T1 (BCR-ABL1 IS ≤ 0.0032%) and T2 (BCR-ABL1 IS > 0.0032%, <1%). Metabolomic profiling was conducted using quadrupole time-of-flight liquid chromatography/mass spectrometry. The data analysis involved a partial least squares discriminant analysis, variable importance in projection (VIP) scores, and a pathway enrichment analysis. Significant metabolites were identified. Results: The PLS-DA revealed distinct metabolomic profiles between CML patients and healthy controls as well as between the T1 and T2 groups. Key differentiating metabolites with VIP scores > 1.5 included glutamate, hypoxanthine, and D-galactonic acid. In the T2 group, significant increases in malate and 5-aminoimidazole-4-carboxamide ribonucleotide were observed, reflecting disruptions in purine metabolism, the tricarboxylic acid cycle, and amino acid metabolism. The pathway enrichment analysis highlighted significant alterations in CML energy metabolism, nucleotide synthesis, and amino acid biosynthesis. Conclusions: CML patients exhibit pronounced metabolic changes, particularly in energy and nucleotide metabolism, which are linked to treatment response. These findings provide novel insights into CML biology and suggest potential biomarkers for monitoring treatment efficacy and predicting outcomes and therapeutic targets for improving treatment outcomes and overcoming tyrosine kinase inhibitor resistance.

## 1. Introduction

Chronic Myeloid Leukemia (CML) is a clonal myeloproliferative neoplasm driven by the BCR::ABL1 fusion gene, formed through the t(9;22) translocation (Philadelphia chromosome) [1]. The BCR::ABL1 fusion gene encodes a constitutively active tyrosine kinase central to CML’s pathogenesis. This aberrant protein activates multiple signaling pathways, including the Janus kinase/signal transducer and the activator of transcription protein 5 (or JAK/STAT5) and mitogen-activated protein kinases/extracellular signal-regulated kinases (or MAPK/ERK), leading to increased cellular proliferation, reduced apoptosis, and altered cellular adhesion. The advent of BCR::ABL1 targeted tyrosine kinase inhibitors (TKIs) and has significantly enhanced clinical outcomes in patients with chronic-phase CML. With TKI therapy, the 5-year overall survival rate now exceeds 90% for individuals diagnosed in the chronic phase [2].

A deep molecular response in CML signifies a log-fold decrease in BCR::ABL1 mRNA from a baseline standardized as the international scale (IS). Milestones include molecular response (MR)4 (≤0.01% IS), MR4.5 (≤0.0032% IS), and MR5 (≤0.001% IS). Given the evidence of an association between the depth of the molecular response and long-term progression-free survival, achieving a deeper MR has become an important goal. Achieving MR4.5 has been consistently associated with improved relapse-free survival following tyrosine kinase inhibitor (TKI) discontinuation across multiple clinical trials. Consequently, current National Comprehensive Cancer Network (NCCN) guidelines designate MR4.5 as a prerequisite criterion for eligibility in TKI cessation [3].

Leukemic cells (LCs) and the bone marrow microenvironment exhibit distinct metabolic properties that influence disease progression and treatment response [4,5]. Altered metabolic pathways, especially in acute leukemias and CML, contribute to pathogenesis, drug resistance, and relapse [6,7]. Glycolysis, fatty acid, and protein metabolism are closely interrelated in LCs. Changes in these metabolisms enable LCs to grow, proliferate, survive, and optimize energy production. The fact that rapidly proliferating cancer cells prefer glycolysis and increase conversion to lactate even in the presence of oxygen due to an increased energy demand is known as the “Warburg Effect”, and this effect is evident in LCs [8]. Fatty acid oxidation and synthesis, essential for energy and membrane biosynthesis, are also elevated [9]. Additionally, oncogenic protein synthesis alters amino acid metabolism, affecting glutamine, serine, glycine, and asparagine [10,11,12]. Despite the success of TKIs, relapses occur due to persistent LCs, drug resistance mutations, and disease progression [13,14]. Understanding metabolic changes can help in developing new therapeutic strategies, improve the efficacy of new and existing therapies, and predict how patients may respond to treatment. Low-molecular-weight metabolites (10–1000 daltons) provide insights into the CML pathogenesis and treatment effects. Real-time metabolic assessments and metabolomic profiling reveal alterations in energy metabolism, biosynthesis, and signaling that support LC survival and proliferation. Metabolic changes in hematologic malignancies remain underexplored, with few studies on treatment response and remission transitions. A study on 29 CML patients found elevated ceramide and reduced sphingomyelin levels in TKI-resistant cases [15]. While BCR::ABL1 mRNA is the primary treatment response biomarker, lipid metabolism variability may offer new options. Reducing the utilization of asparagine by leukemic cells with asparaginase is a good treatment modality in acute lymphoblasticleukemia patients. In a different study, asparaginase was shown to cause apoptosis by depleting glutamine, which is essential for CML cells [16]. Additionally, imatinib reduces cytoplasmic glycolysis in BCR::ABL1+ cells, shifting glucose metabolism toward mitochondrial pathways [17].

This study aims to characterize the metabolomic profile changes associated with CML patients and healthy individuals. Additionally, we aim to delineate alterations within the CML metabolome reflecting transitions into deeper molecular responses, signifying lower residual disease burdens. By integrating metabolomics data stratified by achieving the MR4.5 milestone, the key objectives are to discover stage-specific metabolic signatures in persisting leukemia cells to guide selective therapeutic targeting approaches. Overall, this study will provide a proof-of-concept for the utility of multi-omics interrogation, pairing metabolite profiling with molecular diagnostics to expose response phase-linked metabolic dependencies beyond BCR::ABL1 signaling. This approach not only provides insights into the metabolic underpinnings of the CML progression and treatment response but also has the potential to identify novel therapeutic targets and strategies for overcoming TKI resistance.

## 2. Materials and Methods

### 2.1. Study Design

This study enrolled 51 chronic-phase CML patients (20–76 years) and 24 controls (20–76 years) admitted between February 2024 and June 2024. Patients were grouped as deep molecular responders (MR4.5) (≤0.0032% IS, T1), worse molecular responders (BCR::ABL1 IS > 0.0032%, <1%, T2), and controls. Response to TKI was assessed per European LeukemiaNet guidelines (https://www.leukemia-net.org/, accessed on 1 July 2024) [3]. All the patients have BCR::ABL1 IS < 1%. During the study period, it was not possible to recruit a sufficient number of newly diagnosed patients to allow for meaningful statistical comparisons. Consequently, this patient subgroup was not included in the final analysis. Patients who continued TKI treatment for at least one year and those who switched to a 2nd generation TKI (nilotinib, dasatinib, and bosutinib) due to imatinib intolerance were also included. Accelerated- and blastic-phase CML patients were excluded. In addition, patients with TKI resistance and those who received incomplete doses in the last month to obtain stable plasma drug concentrations were excluded. Patients who had concomitant secondary primary cancer diagnosis, hypertension, diabetes mellitus, hyperlipidemia, hypothyroidism, chronic renal failure, and rheumatologic diseases were also excluded. To minimize other factors influencing metabolites, the groups were matched mainly by age, gender, and body mass index. Informed consent was obtained.

### 2.2. Collection of Plasma Samples

Following an 8–12 h fasting period, blood samples were collected in EDTA-containing hematology tubes. The samples were centrifuged at 3000 rpm for 10 min at 4 °C, and the resulting plasma supernatant was carefully transferred to clean tubes and stored at −80 °C until analysis. On the day of analysis, all samples were transported to the laboratory in dry ice-cooled containers to preserve sample integrity.

### 2.3. Metabolite Extraction

The samples were first thawed at room temperature, after which 0.1 mL of each was transferred into individual tubes. Subsequently, 0.2 mL of a methanol–water solute (7:3, *v*/*v*) was added. The mixture was vortexed for one minute using a vortex mixer (IKA VG 3, Germany) and then left to stand at room temperature for 30 min. Protein precipitation was carried out by centrifugation at 10,000 rpm for 10 min at 4 °C using a refrigerated centrifuge (Hettich Universal 320 R, Tuttlingen, Germany). Following centrifugation, 0.2 mL of the resulting supernatant was carefully collected and dried using a vacuum concentrator (Labconco CentiVap, 7310030, Kansas City, MO, USA). The dried residue was reconstituted in 0.2 mL of an acetonitrile–water mixture (1:1, *v*/*v*), vortexed again for one minute, and centrifuged under the same conditions (10,000 rpm, 10 min, 4 °C). Finally, 0.1 mL of the clear upper layer was transferred into autosampler vials for analysis using a quadrupole time-of-flight liquid chromatography–mass spectrometry (Q-TOF LC/MS) system.

To ensure quality control, pooled samples and extraction blanks were prepared for each group. The metabolite extraction process for all samples was conducted uniformly, and all samples were analyzed under the same conditions.

### 2.4. LC/MS Analyses

Chromatographic separation was carried out using Q-TOF LC/MS (Agilent, 6530, Santa Carla, CA, USA) gradient elution on a reverse-phase column (2.1 × 100 mm, 2.5 µm; XBridge, Waters, Milford, MA, USA). The column temperature was maintained at 35 °C, and the autosampler was kept at 4 °C. A flow rate of 0.4 mL/min was applied. The mobile phases consisted of water (phase A) and acetonitrile (phase B), both containing 0.1% formic acid. The elution gradient was programmed as follows: the run began with 95% phase A, which was decreased to 65% at 2 min, then further reduced to 5% by 8 min. Phase A was returned to 95% by 10 min, followed by a 5 min post-run re-equilibration phase. The injection volume was set at 10 µL.

Mass spectrometric detection was performed in negative ion mode across a mass-to-charge (m/z) range of 75–1200. Sample injections were randomized to minimize systematic bias, and extraction blanks along with quality control samples were injected every six samples to monitor analytical performance and ensure data reliability.

### 2.5. Statistical Analysis

The basic data analysis was performed using IBM SPSS version 23.0. Continuous variables were presented as mean ± standard deviation, while categorical variables were shown as numbers and percentages. To assess the normality of continuous data, we used the Kolmogorov–Smirnov test, and variance homogeneity was evaluated using Levene’s test. For continuous variables that followed a normal distribution, group means were compared using Student’s *t*-test. For non-normally distributed variables, the Mann–Whitney U test was applied. Categorical variables were compared using Pearson’s chi-square test.

The raw chromatographic data was exported from the instrument in “.mzdata” format. For data analysis, several steps were performed using MZmine 2.53 software, including determination of mass-to-charge ratio (or m/z), chromatogram deconvolution, metabolite identification through the in-house library, filtering, and isotope grouping [18]. The in-house library contains 305 endogenous polar and semi-polar metabolites. Metabolites detected in the samples were filtered using extraction blank samples to remove any potential contaminants. The peak areas for each sample were normalized to the average peak area. It expresses a fold change (FC) obtained by dividing the peak area values determined in the chromatogram of each metabolite by the peak area values of the metabolite in different groups. The purpose of calculating these values is to numerically determine the effects on the metabolite signal intensity. These values were evaluated in terms of significance between samples. Metabolites with a fold change ≥1.5 and variable importance in projection (VIP) scores >1.0 above were considered to be significantly affected between groups. In addition, significant metabolites were determined by comparing with independent sample *t*-test (*p* < 0.05). The processed data were further analyzed using MetaboAnalyst 6.0 software [19], which included partial least squares–discriminant analysis (PLS-DA), heat map generation, volcano plot analysis, box plot analysis, and scores.

## 3. Results

This study compared CML patients and healthy controls to identify metabolic alterations specific to CML. By analyzing metabolomic profiles, we aimed to uncover distinct metabolic signatures linked to the CML pathogenesis. Baseline demographic and hematological characteristics were assessed to ensure observed metabolomic differences were not influenced by these factors. The mean age and gender distribution were similar between groups, with unsignificant differences in white blood cell or platelet counts. However, hemoglobin levels were significantly higher in healthy controls (*p* = 0.017), likely due to the inclusion of healthy blood donors (Table 1).

When the CML group was divided into two subgroups based on BCR::ABL1 IS levels (≤0.0032% and >0.0032%, <1%), there were no statistically significant differences between the two groups in terms of the treatment duration, use of TKIs, or EUTOS long-term survival (or ELTS) scores. However, an important difference was observed in the duration of CML, with the T1 group having a longer disease duration than the T2 group (Table 2).

To explore the metabolic differences between healthy controls and CML subgroups, we compared the metabolomic profiles of the control, T1, and T2 groups. The PLS-DA in Figure 1a separates the healthy control group (C, red) and CML subgroups (T1, green, and T2, blue), indicating distinct metabolic profiles for the CML patients compared to healthy individuals. Furthermore, Figure 1b highlights significant differences between the T1 (red) and T2 (green) groups, demonstrating distinct metabolic variations within the CML subgroups. These differences suggest that varying responses to TKI therapy are reflected in the metabolomic profiles of CML patients.

Following the observed distinctions in metabolomic profiles between the healthy control and CML subgroups and between the T1 and T2 groups, we analyzed the data using VIP scores. VIP scores were employed to identify the key metabolites contributing most significantly to these differences. Metabolites with a VIP score greater than 1.5 were considered significant. As shown in Figure 2a, glutamate, hypoxanthine, and D-galactonic acid exhibited the most significant alterations (VIP > 1.8). Additionally, metabolites such as uridine, malate, and 3-hydroxybenzoic acid had VIP scores ranging from 1.5 to 1.8, while alpha-hydroxyisocaproic acid, glycochenodeoxycholic acid, and glycerol-3-phosphate exhibited the least changes (VIP ≤ 1.5). This highlights the specific metabolic pathways that may be key to understanding the disease progression and treatment response in CML patients.

The metabolomic profiles of patients in the T1 and T2 groups displayed significant differences, likely reflecting metabolic changes linked to the TKI response. In the T2 group, the most markedly elevated metabolites were malate and 5-aminoimidazole-4-carboxamide ribonucleotide (AICAR), both with VIP scores greater than 1.8. Other significant metabolites included hypoxanthine, D-galactonic acid, hippurate, citramalate, glutamate, and xanthine (VIP scores between 1.5 and 1.8). The least altered metabolites were 3-(2-Hydroxyphenyl) propanoate and tyrosine, with VIP scores below 1.5 (Figure 2b). These findings suggest that the T2 group experienced more pronounced metabolic shifts compared to both the T1 and healthy control groups, particularly in pathways associated with purine metabolism (hypoxanthine, AICAR, and xanthine), pyrimidine metabolism (uridine), amino acid metabolism (glutamate), the tricarboxylic acid (TCA) cycle (malate), galactose metabolism (D-galactonic acid), and aromatic compound metabolism (3-hydroxybenzoic acid). To further investigate the metabolic shifts between T1 and T2, we generated a box plot (Figure 3) highlighting the most prominent differences between these two groups. These plots illustrate the distribution of normalized metabolite concentrations, allowing the visualization of both the direction and magnitude of the change, as well as the inter-individual variability within each group.

In addition to the VIP scores, heat map visualizations were generated to provide an overview of the relative abundance of metabolites across the healthy control, T1, and T2 groups (Figure 4a). The heat maps allow for the identification of distinct clustering patterns that correspond to specific metabolites. By comparing the heat maps of each group, we can observe how metabolites are upregulated or downregulated in response to CML progression and treatment response. These visualizations enhance our understanding of the metabolic reprogramming occurring in CML, with a clear differentiation between the T1 and T2 subgroups, which may correspond to varying degrees of TKI responses.

To gain deeper insights into the metabolic pathways altered in CML, a pathway analysis was performed comparing the healthy control group with the CML group (Figure 4b). This study identified several significantly altered pathways, particularly those involved in energy metabolism, nucleotide synthesis, and amino acid metabolism. These findings suggest that CML cells undergo significant metabolic reprogramming to support their proliferation and survival, which could serve as potential targets for therapeutic intervention.

## 4. Discussion

A new era has started with TKIs in CML treatment. However, despite these treatments, the rate of patients who do not achieve a molecular response is also considerable. The main aims of our study are to reveal the differences between patients with and without a molecular response at the level of cell biology and metabolic pathways and to reveal new targets to deepen the treatment response. Unlike other cell lines and mouse models, our study utilized an untargeted metabolomic analysis, and many biochemical pathways were evaluated simultaneously in treated patients and healthy controls. The effect of the TKI on the cellular metabolic response was examined in detail. Our results revealed metabolomic differences between CML patients and healthy controls. They also showed differences between CML patients with and without a deep molecular response. These metabolic differences involve multiple pathways, including energy metabolism, nucleotide synthesis, and amino acid metabolism. Figure 1a,b highlight these distinctions, with the PLS-DA showing a clear separation between the healthy control group and both the T1 (deeper response) and T2 (worse response) groups. Additionally, the distinct clustering between T1 and T2 in Figure 1b indicates the presence of metabolomic differences that correspond to treatment efficacy. Figure 2a identifies key metabolites, such as malate, AICAR, and hypoxanthine, with significantly differentiating treatment responses, as shown by their high VIP scores. Figure 3 further emphasizes these differences through a box plot analysis, highlighting statistically significant differences in the metabolite expression between the T1 and T2 groups, identifying key shifts in the metabolic activity in response to the treatment. The heat map in Figure 4a further illustrates the varying metabolite abundances across the healthy control, T1, and T2 groups, supporting the observed metabolic reprogramming. Finally, the enrichment analysis presented in Figure 4b highlights the statistically significant metabolites and their associated pathways between the T1 and T2 groups, indicating that specific metabolic pathways were affected. The analysis of these metabolic pathways suggests a significant metabolic reprogramming in CML cells.

This reprogramming is characterized by an increased nucleotide synthesis and altered energy metabolism. In the T2 group, there was an increase in hypoxanthine, xanthine, AICAR, and uridine levels, which play an active role in DNA repair and DNA and RNA synthesis. The fact that this increase was not detected in the T1 group suggests that it may be a potential therapeutic target to increase the depth of the molecular response in patients with inadequate TKI responses and a high IS score. We also noted an increase in glutamate, which is involved in various cellular processes including protein synthesis, energy metabolism, and neurotransmission. Increased levels of glutamate and malate, especially in the T2 group, suggest changes in mitochondrial function and energy production. Alterations in carbohydrate metabolism are common in cancer cells. Although D-galactonic acid, which is related to galactose metabolism, is not directly involved in cell cycle regulation, it may contribute to altered energy metabolisms. Finally, the metabolite found to be increased in the T2 group was 3-hydroxybenzoic acid, which is involved in aromatic amino acid metabolism and has been associated with an altered gut microbiome.

Hippurate and citramalate increased in the T1 group compared to the T2 group. However, unlike these two metabolites, glycochenodeoxycholic acid and glycerol-3-phosphate increased in the control group compared to the T1 and T2 groups. Since these two separate findings did not support each other, it is difficult to present a clear target for lipid metabolism in the CML cohort.

Since plasma samples were analyzed following protein precipitation, the resulting metabolomic profiles reflect circulating small-molecule metabolites derived from both leukemic and immune cells, as well as systemic metabolic interactions. While this approach does not localize the metabolite origin to a single cell type, it offers valuable insight into the overall metabolic milieu in CML patients under TKI therapy. Notably, several identified metabolites—such as glutamate and AICAR—are known to influence immune cell activation and inflammatory signaling, suggesting a potential link between TKI resistance and immune–metabolic crosstalk.

Our comprehensive metabolomic analysis of the TKI-treated CML cohort revealed a complex landscape of metabolic alterations associated with the disease status and treatment response. This study provides important insights into the metabolic basis of CML progression and TKI efficacy, with potential implications for personalized medicine and therapeutic innovation in hematological malignancies.

### 4.1. Altered Energy Metabolism

Recently, two studies have shown an increased affinity for glucose transporter 1 (or GLUT1) and an increased glucose uptake in CML cells [20,21]. In a study conducted in CML cells obtained from people receiving imatinib treatment, it was shown that imatinib strongly suppressed glycolytic activity, increased the C4-glutamate production from glucose, and activated the mitochondrial TCA cycle (17). Thus, the reduction in anaerobic glycolysis led to the decreased proliferation of CML cells and a decreased glucose demand. This study demonstrated that when low concentrations of imatinib were added to the incubation medium, it reduced the glucose uptake of the cell line. It has been claimed that by using imatinib, the cell can be prevented from using glucose as a substrate and proliferating [17,21,22].

In our study, higher malate levels were found in the T2 group compared to the T1 and healthy control groups. This suggests that the mitochondrial function is impaired in patients with CML who fail to achieve a deep molecular response with the TKI [23]. Malate accumulation indicates a pause in the energy production system of cells due to the impaired function of TCA cycle enzymes. This metabolic change may help leukemia cells to survive despite the TKI treatment. Differences in the energy metabolism between T1 and T2 groups may indicate the ability of CML cells to adapt to treatment. This adaptability is an important factor in the development of resistance to treatment. The ability of CML cells to switch between different metabolic pathways suggests that it will make it difficult to treat these cells effectively. Recent research has shown that CML stem cells have a high energy metabolism, but can be destroyed by oxidative phosphorylation inhibitors, such as metformin [24].

### 4.2. Nucleotide Metabolism Dysregulation

One of the most important results obtained in our study was the changes in nucleotide metabolism. The increase in hypoxanthine and AICAR, especially in the T2 group, provides important information about the nucleotide metabolic status in patients who cannot achieve a deep molecular response with TKI therapy. Hypoxanthine, a purine derivative, serves as a salvage pathway metabolite for purine synthesis. In the T2 group, under the stress of the TKI treatment, cells seem to have developed a different strategy for survival. This strategy is to become more dependent on ways to recycle genetic building blocks called nucleotides (esp. purine). This is in line with the findings of Nathanson et al. who showed that the metabolism of genetic building blocks is important for the survival of leukemia stem cells [25]. Of particular interest are the higher levels of a molecule called AICAR in the T2 group. AICAR activates AMP-activated protein kinase (or AMPK), an important protein involved in the energy balance of cells. AMP-activated protein kinase is known to promote cancer cell survival under metabolic stress [26]. The fact that AICAR is elevated in patients who fail to achieve a deep molecular response with TKIs suggests that cells may be endeavoring to survive through AMP-activated protein kinase. These findings provide opportunities for new combination therapies targeting both the leukemia-causing BCR-ABL protein and the metabolic vulnerabilities of cells.

### 4.3. Amino Acid Metabolism

In our study, glutamate, which is involved in the amino group transfer and energy production by converting to α-ketoglutarate in the TCA cycle, was found to be higher in the T2 group compared to both T1 and healthy control groups. Glutamine dependence is a well-known feature of many cancers, including leukemias [27]. The higher glutamate levels observed in the T2 group may reflect increased glutaminolysis, providing energy and biosynthetic precursors for cell survival. In studies conducted in acute myeloid leukemia patients, glutamine, the amidated form of glutamate, was found to be low compared to healthy controls. It has been claimed that glutamine is an important protein source, especially for rapidly proliferating leukemic cells, and its plasma levels are lower than controls due to its use in the biosynthesis of purines and pyrimidines involved in DNA and RNA synthesis [22,28]. According to the data obtained in our study, the glutamine-to-glutamate turnover increased in CML cells. In light of previous studies, the inhibition of glutamate conversion as a new target and the depletion of glutamine may be associated with increased mitochondrial oxidative stress, which results in the death of CML cells [29].

Glutamine is an essential factor for the immune system. However, targeting glutamine metabolism is considered a strategy to improve TKI efficacy in CML patients. It should be noted that this strategy may impair the anticancer effects of immune cells in the bone marrow microenvironment and thus the antileukemic effect. Therefore, the biggest challenge in future studies will be targeting glutamine metabolism in cancer cells without harming the immune system.

### 4.4. Lipid Metabolism

The varying glycerol-3 phosphate levels between the healthy control and T1 and T2 groups indicate differences in lipid metabolism in CML patients. Recent studies have emphasized the importance of lipid metabolism in hematological malignancies. Ye et al. have shown that cytokines secreted by adipose tissue increased adipose tissue lipolysis and the free fatty acid release in CML cell lines and mouse models. This is evidence of increased fatty acid oxidation in CML cells. In addition, in this study, an increase in the FAO ratio was shown in CD36+ drug-resistant CML cells, and it was claimed that it may be related to drug resistance [5]. As a mechanism, it is thought that a fatty acid oxidation-induced increase in cellular nicotinamide adenine dinucleotide phosphate (or NADPH) and consequently a decrease in cellular oxidative stress may lead to the prevention of apoptosis in CML cells [30,31]. In another study, a fatty acid oxidation inhibition resulted in increased apoptosis in leukemic cells, supporting this claim [32]. In our study, an increase in glycerol-3 phosphate was found only in the control arm. It could not clearly distinguish between T1 and T2 groups, but it was shown to be generally decreased in CML patients compared to healthy subjects. A study with a similar design to our study showed a significant difference in glycerol levels between TKI-sensitive CML patients and TKI-resistant CML patients [33]. The differential regulation of lipid-related metabolites between TKI-responsive and -resistant patients warrants further investigation as it may reveal new therapeutic targets or biomarkers of treatment response.

### 4.5. Novel Metabolic Signatures

The high levels of D-galactonic acid observed in the T2 group represent a novel finding in CML metabolomics. Although the specific role of D-galactonic acid in CML biology is not yet clear, these changes in D-galactonic acid metabolism may reflect changes in the cellular redox status or pentose phosphate pathway activity. This observation is in concordance with the study by Patra and Hay, which emphasizes the importance of the pentose phosphate pathway in cancer metabolism and redox balance [34]. When Yang et al. compared the metabolic biomarkers of newly diagnosed CML patients with healthy controls, it was shown that carbohydrate (D-galactose) levels increased and decreased again with treatment in parallel with the results in our study [33].

The low levels of hippurate and citramalate observed in the T2 group are interesting findings that were not prominent in previous CML studies. Hippurate, a conjugate of glycine and benzoic acid, is mainly obtained from the liver by metabolizing benzoic acid. It has clinical data to evaluate liver reserve, especially in hepatocellular carcinoma. In addition, urinary hippurate levels have been shown to decrease in some malignancies. The fact that it is found to be low only in patients without a molecular response, independent of TKI use, warrants a further investigation for its use as a potential marker in disease monitoring. There is no clear data on the association of citramalate, an intermediate in isoleucine synthesis, with cancer.

This study offers a comprehensive metabolomic analysis of CML patients undergoing TKI therapy, unveiling significant metabolic alterations associated with disease progression and treatment response. We identified key metabolites and disrupted pathways that distinguish these groups by comparing the metabolomic profiles of healthy controls and CML patients stratified into T1 and T2 groups. Our findings highlight substantial changes in energy metabolism, nucleotide synthesis, and amino acid metabolism in CML patients, especially those with suboptimal responses to TKI therapy. Elevated levels of metabolites such as malate, AICAR, hypoxanthine, and glutamate in the T2 group suggest that leukemic cells may adopt alternative metabolic pathways to sustain proliferation and survival despite TKI treatments. These metabolic adaptations could contribute to treatment resistance and disease persistence. Identifying these specific metabolic alterations provides valuable insights into the underlying biology of CML and suggests potential metabolic vulnerabilities that could be therapeutically targeted. For instance, interventions aimed at disrupting purine metabolism, glutaminolysis, or mitochondrial function may enhance the efficacy of existing TKIs and help overcome resistance. Future studies should focus on validating these metabolic markers in more extensive, independent cohorts and investigating their potential as prognostic indicators or therapeutic targets. Additionally, integrating metabolomic data with other omics approaches—such as genomics, proteomics, and transcriptomics—could provide a more comprehensive understanding of CML pathogenesis and treatment responses. Such multi-omics strategies may facilitate the development of personalized treatment plans and novel therapeutic agents that improve patient outcomes.

### 4.6. Limitations

This study has several limitations. Firstly, in this study, an untargeted metabolomic approach was initially employed to derive conclusive insights into individual metabolic pathways. Metabolite profiling provides opportunities for the identification of biomarkers and the stratification of patients for personalized therapeutic interventions. While targeted metabolomics is utilized to test specific hypotheses or to monitor selected metabolites, untargeted analyses aim to detect the full spectrum of metabolites present in a given sample. Consequently, targeted metabolomic analyses may offer more specific and clinically actionable results in the context of personalized medicine. Secondly, while our study population is representative, it is still relatively small, which limits the statistical power to make definitive conclusions about the metabolic differences between groups. Therefore, a subgroup analysis among TKIs could not be performed. One of the other limitations of our study is the absence of newly diagnosed, treatment-naive CML patients. This was due to the difficulty in recruiting a sufficient number of eligible patients within the study period. As a result, all patients included in the metabolomic analysis had been receiving TKI therapy for at least one year. While this limits our ability to characterize the baseline (pre-treatment) metabolomic landscape of CML, it also provides an advantage in that it allows for a controlled comparison between molecular responders and non-responders within a homogenous treatment context. Although TKI therapy is known to influence cellular metabolism, we minimized the confounding effects of the treatment variability by including only chronic-phase patients with a stable TKI exposure and by excluding individuals with resistance, recent dose changes, or treatment interruptions. Importantly, since all patients had been receiving treatment, the observed metabolic differences between T1 (deep responders) and T2 (suboptimal responders) groups are more likely to reflect differences in biological responses to therapy, rather than the effects of the therapy itself. Future studies including treatment-naive patients will be necessary to delineate disease-intrinsic metabolic alterations and to better isolate the direct impact of TKI therapy on the metabolome. Therefore, the metabolomic profiling of bone marrow could not also be performed in this study. This situation prevented the evaluation of metabolic profiles in every period of CML disease from diagnosis to the molecular response. In addition, since all of the patients included in this study were cytogenetically responsive, leukocyte samples were not studied in blood samples. Future studies with larger cohorts are necessary to validate these findings and explore the potential of metabolomic profiling in CML further.

## 5. Conclusions

Our findings highlight the pivotal role of metabolic reprogramming in the progression of CML and the development of treatment resistance. By elucidating the specific metabolic pathways altered in response to TKI therapy, we pave the way for new avenues in CML research and treatment. Targeting these metabolic pathways could enhance the efficacy of existing therapies, overcome resistance, and ultimately improve long-term outcomes for patients with CML.

## Figures and Tables

**Figure 1 metabolites-15-00376-f001:**
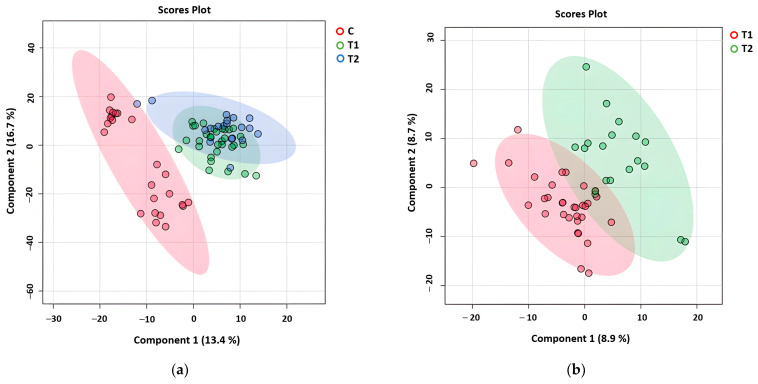
A PLS-DA plot showing the separation of metabolomic profiles across (**a**) the healthy control group (C, red), deeper molecular response group (T1, green), and worse molecular response group (T2, blue), (**b**) T1 (red) and T2 (green).

**Figure 2 metabolites-15-00376-f002:**
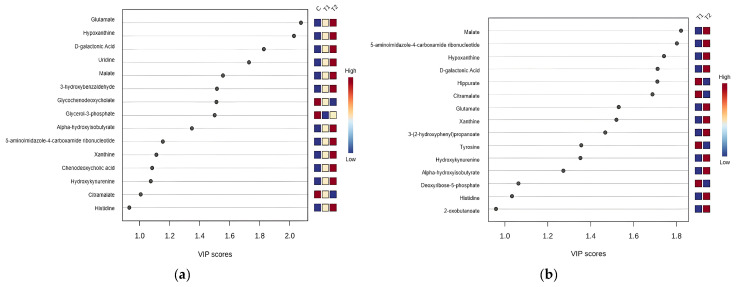
VIP score plots identifying the metabolites that contribute most significantly to the differentiation between (**a**) the healthy control, deeper molecular response (T1), and worse molecular response (T2) groups and (**b**) the T1 and T2 groups. Metabolites with VIP scores greater than 1.5 were considered significant. Metabolites such as malate and AICAR have the highest VIP scores, indicating significant changes in the T2 group.

**Figure 3 metabolites-15-00376-f003:**
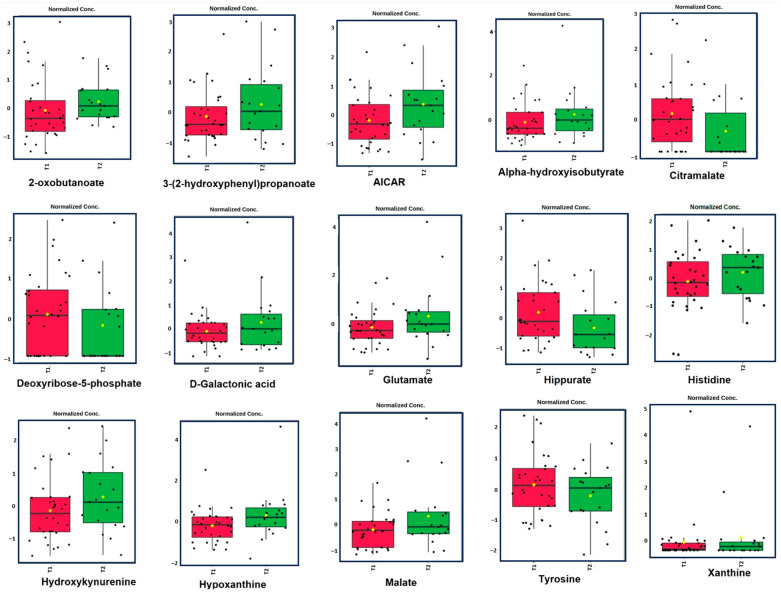
Box plots showing normalized concentrations of selected discriminatory metabolites between the T1 (red) and T2 (green) groups.

**Figure 4 metabolites-15-00376-f004:**
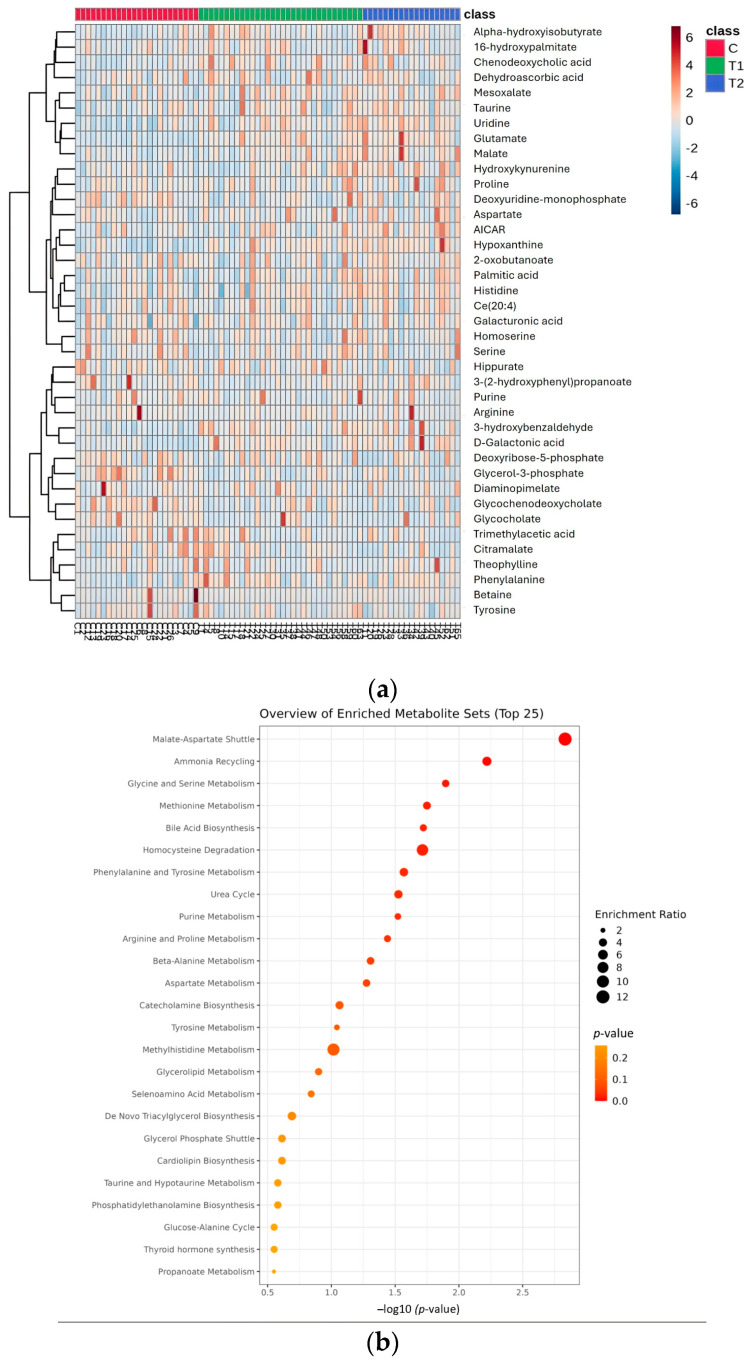
(**a**) The heat map representing the relative abundance of metabolites across the healthy control (C, red), deeper molecular response (T1 green), and worse molecular response (T2, blue) groups. (**b**) The pathway enrichment analysis comparing the healthy control and CML groups. The plot highlights the most significantly enriched metabolic pathways, with larger circles representing a higher pathway impact and darker colors indicating lower *p*-values (greater statistical significance).

**Table 1 metabolites-15-00376-t001:** Demographic characteristics and laboratory data of the CML and healthy control groups.

	Control (n = 24)	CML (n = 51)	*p*-Value
Mean Age (years) ± SD	48.7 (±7.8)	51.1 (±14.8)	0.460
Gender (n) (%)			
Male	13 (54.2%)	31 (60.8%)	0.487
Female	11 (45.8%)	20 (39.2%)	
Mean WBC (mm)3 ±SD	7204.6 (±1649.1)	6781.8 (±1934.0)	0.374
Mean HGB (gr/dL) ±SD	14.2 (±1.1)	13.5 (±1.2)	0.017
Mean PLT (mm)3 ±SD	261,827.5 (±54,621.2)	250,745.1 (±101,660.4)	0.656

CML: Chronic Myeloid Leukemia, WBC: white blood cell, HGB: hemoglobin, PLTs: platelets, and SD: standard deviation.

**Table 2 metabolites-15-00376-t002:** CML group (T1, T2) according to BCR::ABL1 IS level, duration of treatment, medications used, risk classifications, and BCR::ABL1 IS levels at diagnosis.

	T1	T2	*p*-Value
BCR::ABL1 IS Level (n) (%)	32 (≤0.0032)	19 (>0.0032, <1)	
BCR::ABL1 IS level at diagnosis (%)	92.9 ± 106.2	84.8 ± 64.6	0.707
Mean Duration of CML (year) ±SD	9.1 ± 6.2	3.1 ± 2.4	0.000
Line of Treatment (n) (%)			0.472
First	21 (65.6)	14 (73.7)	
Second	7 (21.9)	3 (15.8)	
Third	4 (12.5)	2 (10.5)	
TKI (n) (%)			0.76
Imatinib	22 (68.8)	13 (68.4)	
Dasatinib	4 (12.5)	2 (10.5)	
Nilotinib	5 (15.6)	3 (15.8)	
Bosutinib	1 (3.1)	1 (5.3)	
Risk Category			0.69
(ELTS score) (n) (%)			
Low	4 (12.5)	2 (10.5)	
Intermediate	19 (59.4)	10 (52.6)	
High	9 (28.1)	7 (36.9)	

CML: Chronic Myeloid Leukemia, TKI: tyrosine kinase inhibitor, and ELTS: EUTOS long-term survival score.

## Data Availability

The data supporting the findings of this study are not publicly available due to privacy and confidentiality considerations but may be obtained from the corresponding author upon reasonable request.

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
