# Peer review of "Metabolomic Profiling and Bioanalysis of Chronic Myeloid Leukemia: Identifying Biomarkers for Treatment Response and Disease Monitoring"

_metabolites, 2025, doi:10.3390/metabo15060376_

Round 1
Reviewer 1 Report
Comments and Suggestions for Authors
The work examined changes in the metabolome in patients with CML and obtained interesting results. However, a number of corrections are required:
Line 108 - "Newly diagnosed CML patients were not included in the study since sufficient numbers could not be reached." - It is unclear what the authors wanted to say - that a group of primary patients was not recruited or that they might not have achieved remission?
Figure 1a looks like a duplicate of figure 1b - there is no worse molecular response group (T2, blue) on the graph; in the legend "a" there are T1 (red) and T2 (green), in the caption to the figure - the healthy control group (C, red), deeper molecular response group (T1, green), and worse molecular response group (T2, blue)
The caption to Figure 2 indicates the colors for each group, but in the figure they are either not used or not visible.
Figure 3 would be greatly improved if the most altered metabolites, such as malate and AICAR, were highlighted.
Since the study could not be performed in primary patients, it is necessary to discuss at the outset how TKI therapy may have influenced or why it should not have influenced the results obtained.
Author Response
Dear Reviewer;
First of all, thank you very much for your contributions that make our article more valuable.
We made corrections in line with your requests according to your Major and Minor comments below. We tried to Response your questions as much as possible.
Reviewer-1: The work examined changes in the metabolome in patients with CML and obtained interesting results. However, a number of corrections are required:
Comments-1.1: Line 108 - "Newly diagnosed CML patients were not included in the study since sufficient numbers could not be reached." - It is unclear what the authors wanted to say - that a group of primary patients was not recruited or that they might not have achieved remission?
Response-1.1: We thank the reviewer for this important comment and the opportunity to clarify our intention. The primary aim of our study was to investigate the metabolic differences between CML patients with deep molecular response (MR4.5) and those with suboptimal response under tyrosine kinase inhibitor (TKI) therapy. Ideally, we had intended to include a third group of newly diagnosed, treatment-naïve CML patients in order to capture the baseline metabolic state before any TKI exposure. However, during the study period, we were unable to recruit a sufficient number of such patients to ensure meaningful statistical comparison. As a result, we focused our analysis on patients who had been on TKI therapy and achieved varying levels of molecular response. We agree that including newly diagnosed, untreated patients could have enriched the study by providing a clearer understanding of how TKI therapy alters the metabolic profile from the baseline state. This would allow the identification of therapy-induced versus disease-inherent metabolic changes more precisely. We have revised the manuscript accordingly and explicitly acknowledged this point in the Limitations section. The updated text now reads: “During the study period, it was not possible to recruit a sufficient number of newly diagnosed patients to allow for meaningful statistical comparisons. Consequently, this patient subgroup was not included in the final analysis.’’
Comments-1.2: Figure 1a looks like a duplicate of figure 1b - there is no worse molecular response group (T2, blue) on the graph; in the legend "a" there are T1 (red) and T2 (green), in the caption to the figure - the healthy control group (C, red), deeper molecular response group (T1, green), and worse molecular response group (T2, blue)
Response-1.2: We sincerely thank the reviewer for pointing out this important issue. You are absolutely correct—Figure 1a was inadvertently duplicated during the formatting and editing process when adapting the manuscript to the journal’s template. This was a copy/paste error, and we apologize for the oversight. The correct version of Figure 1b has been added. We would like to emphasize that the text describing the findings related to Figure ‘’1a’’ an ‘’1b’’ is accurate.
Comments-1.3: The caption to Figure 2 indicates the colors for each group, but in the figure they are either not used or not visible.
Response-1.3: You are absolutely right—the original figure caption included references to group colors (e.g., red, green, blue), which were not reflected in the actual figure. These color designations were initially adapted from the PCA figure and mistakenly retained, although they were not necessary for interpreting the VIP score plot. Upon reviewing this, we recognized it added confusion rather than clarity. As suggested, we have removed the color references from the caption in the revised version of the manuscript.
The revised caption for Figure 2 now reads:
Figure 2.
VIP score plots identifying the metabolites that contribute most significantly to the differentiation between (a) the healthy control, deeper molecular response (T1), and worse molecular response (T2) groups, and (b) the T1 and T2 groups. Metabolites with VIP scores greater than 1.5 were considered significant. Metabolites such as malate and AICAR have the highest VIP scores, indicating significant changes in the T2 group.
Comments-1.4: Figure 3 would be greatly improved if the most altered metabolites, such as malate and AICAR, were highlighted.
Response-1.4: We thank the reviewer for the helpful suggestion regarding visualization of key metabolites such as malate and AICAR. In line with your comment, we re-evaluated the most informative and transparent way to present metabolite-level differences between the T1 and T2 groups. Rather than modifying the volcano plot, we chose to replace it with a panel of box plots (now Figure 3) depicting the normalized concentrations of selected discriminatory metabolites across both groups. This figure presents not only the direction and magnitude of change but also individual data points, medians, and distribution spread—providing a clearer and more accurate reflection of within-group variability. We found that volcano plots, although visually intuitive, do not capture biological variation between individuals and may suggest misleading precision, as they lack error estimates such as confidence intervals or standard deviations. In contrast, the box plot format offers a more granular, biologically informative view by visualizing the full data distribution per metabolite, which we believe aligns better with the study’s clinical and translational goals.
Comments-1.5: Since the study could not be performed in primary patients, it is necessary to discuss at the outset how TKI therapy may have influenced or why it should not have influenced the results obtained.
Response-1.5: As previously noted in our response to Comment #1, we had initially planned to include newly diagnosed, treatment-naïve CML patients to better characterize baseline metabolic profiles. However, due to limited recruitment during the study period, the number of such patients was insufficient for meaningful comparison, and therefore they were excluded from the analysis. We have acknowledged this as a study limitation in the revised manuscript. Regarding the potential influence of TKI therapy on the observed metabolomic alterations, we fully agree that treatment may significantly impact metabolic profiles. In fact, this was one of the central hypotheses of our study—to investigate how metabolomic signatures differ among CML patients with varying degrees of response under long-term TKI therapy. Our study design, which compares patients achieving deep molecular response (T1) with those showing suboptimal response (T2) under TKI exposure, provides an opportunity to evaluate treatment-associated metabolic adaptation. While we cannot completely dissociate disease-related from drug-induced effects due to the lack of pre-treatment samples, the differential metabolomic profiles observed between T1 and T2 suggest that the metabolic phenotype may reflect both residual disease activity and the cellular response to therapy. Therefore, we interpret our findings as capturing the composite metabolic state under TKI pressure, which could represent a combination of intrinsic leukemic cell biology, therapy-induced adaptation, and microenvironmental interactions. Future studies incorporating longitudinal sampling—before and after therapy—are essential to fully disentangle these overlapping effects. We have added this point to the revised limitation section to provide appropriate context. One of the major limitations of our study is the absence of newly diagnosed, treatment-naïve CML patients. This was due to the difficulty in recruiting a sufficient number of eligible patients within the study period. As a result, all patients included in the metabolomic analysis had been receiving TKI therapy for at least one year. While this limits our ability to characterize the baseline (pre-treatment) metabolomic landscape of CML, it also provides an advantage in that it allows for a controlled comparison between molecular responders and non-responders within a homogenous treatment context. Although TKI therapy is known to influence cellular metabolism, we minimized the confounding effects of treatment variability by including only chronic-phase patients with stable TKI exposure and by excluding individuals with resistance, recent dose changes, or treatment interruptions. Importantly, since all patients had been under treatment, the observed metabolic differences between T1 (deep responders) and T2 (suboptimal responders) groups are more likely to reflect differences in biological response to therapy, rather than the effects of therapy itself. Future studies including treatment-naïve patients will be necessary to delineate disease-intrinsic metabolic alterations and to better isolate the direct impact of TKI therapy on the metabolome.

Reviewer 2 Report
Comments and Suggestions for Authors
This is a novel study investigating treatment-associated metabolic changes in CML patients compared to healthy controls, leading to the identification of potential metabolic biomarkers for distinguishing treatment responses among CML patients. The manuscript is well-written and easy to understand. I have the following comments:
Figure 1a: There is an error in this figure; the wrong plot appears to have been used for panel 1a. Please correct this.
Figure 3: Please indicate the locations of malate, AICAR, and other significantly altered metabolites in the plot. Also, clarify what "FC" represents—does it refer to fold change or another metric?
Summary and Network Figures: Please provide an additional figure summarizing the key findings of this study and their clinical relevance. More importantly, include an interactive network diagram of TKI-resistant metabolites to clearly present the major findings. This diagram will be valuable for translational applications, particularly in designing future strategies to predict and prevent CML progression.
Discussion:
Given that plasma samples were used in this study, please discuss whether the observed metabolic changes originate from leukemic cells, immune cells, or both. Additionally, elaborate on how these metabolites might influence CML-related immune responses.
Limitations:
Regarding the first limitation mentioned, please rephrase it for clarity so that the underlying point is more explicitly conveyed.
Author Response
Dear Reviewer-2;
First of all, thank you very much for your contributions that make our article more valuable.
In line with your suggestions; We made corrections in line with your requests according to your Major and Minor comments below. We tried to Response your questions as much as possible.
Reviewer-2: This is a novel study investigating treatment-associated metabolic changes in CML patients compared to healthy controls, leading to the identification of potential metabolic biomarkers for distinguishing treatment responses among CML patients. The manuscript is well-written and easy to understand. I have the following comments:
Comments-1.1: Figure 1a: There is an error in this figure; the wrong plot appears to have been used for panel 1a. Please correct this.
Response-1.1: You are absolutely correct—Figure 1a was inadvertently duplicated during the formatting and editing process when adapting the manuscript to the journal’s template. This was a copy/paste error, and we apologize for the oversight. We would like to emphasize that the text describing the findings related to Figure ‘’1a’’ and ‘’1b’’ is accurate. Specifically, as stated: “PLS-DA plot showing the separation of metabolomic profiles between (a) the healthy control group (C, red), deeper molecular response group (T1, green), and worse molecular response group (T2, blue), (b) T1 (red) and T2 (green)’’.
Comments-1.2: Figure 3: Please indicate the locations of malate, AICAR, and other significantly altered metabolites in the plot. Also, clarify what "FC" represents—does it refer to fold change or another metric?
Response-1.2: We thank the reviewer for the helpful suggestion regarding visualization of key metabolites such as malate and AICAR. In line with your comment, we re-evaluated the most informative and transparent way to present metabolite-level differences between the T1 and T2 groups. Rather than modifying the volcano plot, we chose to replace it with a panel of box plots (now Figure 3) depicting the normalized concentrations of selected discriminatory metabolites across both groups. This figure presents not only the direction and magnitude of change but also individual data points, medians, and distribution spread—providing a clearer and more accurate reflection of within-group variability. We found that volcano plots, although visually intuitive, do not capture biological variation between individuals and may suggest misleading precision, as they lack error estimates such as confidence intervals or standard deviations. In contrast, the box plot format offers a more granular, biologically informative view by visualizing the full data distribution per metabolite, which we believe aligns better with the study’s clinical and translational goals.
Comments-1.3: Summary and Network Figures: Please provide an additional figure summarizing the key findings of this study and their clinical relevance. More importantly, include an interactive network diagram of TKI-resistant metabolites to clearly present the major findings. This diagram will be valuable for translational applications, particularly in designing future strategies to predict and prevent CML progression.
Response-1.3: A graphical abstract summarizing the study design and findings attached in main manuscript and also, uploaded as a additional file.
Comments-1.4: Given that plasma samples were used in this study, please discuss whether the observed metabolic changes originate from leukemic cells, immune cells, or both. Additionally, elaborate on how these metabolites might influence CML-related immune responses.
Response-1.4: We appreciate this insightful comment. In our study, peripheral blood plasma was collected using EDTA tubes and processed via centrifugation to separate the cell-free fraction. During sample preparation, plasma proteins were precipitated using a cold methanol-based extraction protocol. This approach eliminates high-abundance proteins—such as albumin and immunoglobulins—and leaves behind a metabolite-rich supernatant composed primarily of small molecules (<1.5 kDa), including amino acids, nucleotides, organic acids, and other intermediates. These metabolites represent the systemic metabolic state resulting from a combination of cellular activities, including those of leukemic cells, immune cells, stromal components, and metabolic interactions in the bone marrow and peripheral circulation.
Although plasma metabolomics does not allow direct attribution of each metabolite to a specific cell type, the metabolic signature we observe reflects cumulative inputs from both leukemic and host immune cells, particularly under TKI therapy. For example, elevated levels of AICAR and glutamate in poor responders (T2 group) may be associated with altered energy metabolism and purine salvage in leukemic cells, while also potentially modulating immune cell activation, redox state, and cytokine signaling. Several of the altered metabolites identified—such as glutamate, malate, and hypoxanthine—are known to influence immune function. Glutamate, for instance, can regulate T-cell activation and proliferation; AICAR is a modulator of AMP-activated protein kinase (AMPK), which affects inflammatory responses; and changes in nucleotide metabolism may reflect heightened turnover and stress in both leukemic and immune cell populations. In the revised manuscript, we have expanded the Discussion to include this explanation and the following paragraph has been added: “Since plasma samples were analyzed following protein precipitation, the resulting metabolomic profiles reflect circulating small-molecule metabolites derived from both leukemic and immune cells, as well as systemic metabolic interactions. While this approach does not localize metabolite origin to a single cell type, it offers valuable insight into the overall metabolic milieu in CML patients under TKI therapy. Notably, several identified metabolites—such as glutamate and AICAR—are known to influence immune cell activation and inflammatory signaling, suggesting a potential link between TKI resistance and immune-metabolic crosstalk.”
Comments-1.5: Limitations: Regarding the first limitation mentioned, please rephrase it for clarity so that the underlying point is more explicitly conveyed.
Response-1.5: The first limitations was explained in detail. “Firstly, in this study, an untargeted metabolomic approach was initially employed to derive conclusive insights into individual metabolic pathways. Metabolite profiling provides opportunities for the identification of biomarkers and the stratification of patients for personalized therapeutic interventions. While targeted metabolomics is utilized to test specific hypotheses or to monitor selected metabolites, untargeted analyses aim to detect the full spectrum of metabolites present in a given sample. Consequently, targeted metabolomic analyses may offer more specific and clinically actionable re-sults in the context of personalized medicine”.

Round 2
Reviewer 1 Report
Comments and Suggestions for Authors
The authors have corrected inaccuracies and taken into account all comments.